# Learning and Inference in Hilbert Space with Quantum Graphical Models

**Siddarth Srinivasan**
College of Computing
Georgia Tech
Atlanta, GA 30332
sidsrini@gatech.edu

**Carlton Downey**
Department of Machine Learning
Carnegie Mellon University
Pittsburgh, PA 15213
cmdowney@cs.cmu.edu

**Byron Boots**
College of Computing
Georgia Tech
Atlanta, GA 30332
bboots@cc.gatech.edu

## Abstract

Quantum Graphical Models (QGMs) generalize classical graphical models by adopting the formalism for reasoning about uncertainty from quantum mechanics. Unlike classical graphical models, QGMs represent uncertainty with density matrices in complex Hilbert spaces. Hilbert space embeddings (HSEs) also generalize Bayesian inference in Hilbert spaces. We investigate the link between QGMs and HSEs and show that the sum rule and Bayes rule for QGMs are equivalent to the kernel sum rule in HSEs and a special case of Nadaraya-Watson kernel regression, respectively. We show that these operations can be kernelized, and use these insights to propose a Hilbert Space Embedding of Hidden Quantum Markov Models (HSE-HQMM) to model dynamics. We present experimental results showing that HSE-HQMMs are competitive with state-of-the-art models like LSTMs and PSRNNs on several datasets, while also providing a nonparametric method for maintaining a probability distribution over continuous-valued features.

## 1   Introduction and Related Work

Various formulations of Quantum Graphical Models (QGMs) have been proposed by researchers in physics and machine learning [Srinivasan et al., 2018, Yeang, 2010, Leifer and Poulin, 2008] as a way of generalizing probabilistic inference on graphical models by adopting quantum mechanics' formalism for reasoning about uncertainty. While Srinivasan et al. [2018] focused on modeling dynamical systems with Hidden Quantum Markov Models (HQMMs) [Monras et al., 2010], they also describe the basic operations on general quantum graphical models, which generalize Bayesian reasoning within a framework consistent with quantum mechanical principles. Inference using Hilbert space embeddings (HSE) is also a generalization of Bayesian reasoning, where data is mapped to a Hilbert space in which kernel sum, chain, and Bayes rules can be used [Smola et al., 2007, Song et al., 2009, 2013]. These methods can model dynamical systems such as HSE-HMMs [Song et al., 2010], HSE-PSRs [Boots et al., 2012], and PSRNNs [Downey et al., 2017]. [Schuld and Killoran, 2018] present related but orthogonal work connecting kernels, Hilbert spaces, and quantum computing.

Since quantum states live in complex Hilbert spaces, and both QGMs and HSEs generalize Bayesian reasoning, it is natural to ask: what is the relationship between quantum graphical models and Hilbert space embeddings? This is precisely the question we tackle in this paper. Overall, we present four contributions: (1) we show that the sum rule for QGMs is identical to the kernel sum rule for HSEs, while the Bayesian update in QGMs is equivalent to performing Nadaraya-Watson kernel regression, (2) we show how to kernelize these operations and argue that with the right choice of features, we are mapping our data to quantum systems and modeling dynamics as quantum state evolution, (3) we use these insights to propose a HSE-HQMM to model dynamics by mapping data to quantum systems and performing inference in Hilbert space, and, finally, (4) we present a learning algorithm and experimental results showing that HSE-HQMMs are competitive with other state-of-the-art methods for modeling sequences, while also providing a nonparametric method for estimating the distribution of continuous-valued features.

## 2 Quantum Graphical Models

### 2.1 Classical vs Quantum Probability

In classical discrete graphical models, an observer's uncertainty about a random variable $X$ can be represented by a vector $\vec{x}$ whose entries give the probability of $X$ being in various states. In quantum mechanics, we write the 'pure' quantum state of a particle $A$ as $|\psi\rangle_A$, a complex-valued column-vector in some orthonormal basis that lives in a Hilbert space, whose entries are 'probability amplitudes' of system states. The squared norm of these probability amplitudes gives the probability of the corresponding system state, so the sum of squared norms of the entries must be 1. To describe 'mixed states', where we have a probabilistic mixture of quantum states, (e.g. a mixture of $N$ quantum systems, each with probability $p_i$) we use a Hermitian 'density matrix', defined as follows:

$$\hat{\rho} = \sum_i^N p_i |\psi_i\rangle\langle\psi_i| \tag{1}$$

The diagonal entries of a density matrix give the probabilities of being in each system state, and off-diagonal elements represent quantum coherences, which have no classical interpretation. Consequently, the normalization condition is $\text{tr}(\hat{\rho}) = 1$. Uncertainty about an $n$-state system is represented by an $n \times n$ density matrix. The density matrix is the quantum analogue of the classical belief $\vec{x}$.

### 2.2 Operations on Quantum Graphical Models

Here, we further develop the operations on QGMs introduced by Srinivasan et al. [2018], working with the notion that the density matrix is the quantum analogue of a classical belief state.

**Joint Distributions** The joint distribution of an $n$-state variable $A$ and $m$-state variable $B$ can be written as an $nm \times nm$ 'joint density matrix' $\hat{\rho}_{AB}$. When $A$ and $B$ are independent, $\hat{\rho}_{AB} = \hat{\rho}_A \otimes \hat{\rho}_B$. As a valid density matrix, the diagonal elements represent probabilities corresponding to the states in the Cartesian product of the basis states of the composite variables (so $\text{tr}(\hat{\rho}_{AB}) = 1$).

**Marginalization** Given a joint density matrix, we can recover the marginal 'reduced density matrix' for a subsystem of interest with the 'partial trace' operation. This operation is the quantum analogue of classical marginalization. For example, the partial trace for a two-variable joint system $\hat{\rho}_{AB}$ where we trace over the second particle to obtain the state of the first particle is:

$$\hat{\rho}_A = \text{tr}_B\left(\hat{\rho}_{AB}\right) = \sum_j {}_B\langle j|\hat{\rho}_{AB}|j\rangle_B \tag{2}$$

Finally, we discuss the quantum analogues of the sum rule and Bayes rule. Consider a prior $\vec{\pi} = P(X)$ and a likelihood $P(Y|X)$ represented by the column stochastic matrix $\mathbf{A}$. We can then ask two questions: what are $P(Y)$ and $P(X|y)$?

**Sum Rule** The classical answer to the first question involves multiplying the likelihood with the prior and marginalizing out $X$, like so:

$$P(Y) = \sum_x P(Y|x)P(x) = \mathbf{A}\vec{\pi} \tag{3}$$

Srinivasan et al. [2018] show how we can construct a quantum circuit to perform the classical sum rule (illustrated in Figure 1a, see appendix for note on interpreting quantum circuits). First, recall that all operations on quantum states must be represented by unitary matrices in order to preserve the 2-norm of the state. $\hat{\rho}_{env}$ is an environment 'particle' always prepared in the same state that will eventually encode $\hat{\rho}_Y$: it is initially a matrix with zeros everywhere except $\hat{\rho}_{1,1} = 1$. Then, if the prior $\vec{\pi}$ is encoded in a density matrix $\hat{\rho}_X$, and the likelihood table $\mathbf{A}$ is encoded in a higher-dimensional unitary matrix, we can replicate the classical sum rule. Letting the prior $\hat{\rho}_X$ be *any* density matrix and $\hat{U}_1$ be any unitary matrix generalizes the circuit to perform the 'quantum sum rule'. This circuit can be written as the following operation (the unitary matrix produces the joint distribution, the partial trace carries out the marginalization):

$$\hat{\rho}_Y = \text{tr}_X\left(\hat{U}_1\left(\hat{\rho}_X \otimes \hat{\rho}_{env}\right)\hat{U}_1^\dagger\right) \tag{4}$$

**Bayes Rule** Classically, we perform Bayesian update as follows (where $\text{diag}(\mathbf{A}_{(:,y)})$ selects the row of matrix $\mathbf{A}$ corresponding to observation $y$ and stacks it along a diagonal):

$$P(X|y) = \frac{P(y|X)P(X)}{\sum_x(y|x)P(x)} = \frac{\text{diag}(\mathbf{A}_{(y,:)})\vec{\pi}}{\mathbb{1}^T\text{diag}(\mathbf{A}_{(y,:)})\vec{\pi}} \tag{5}$$

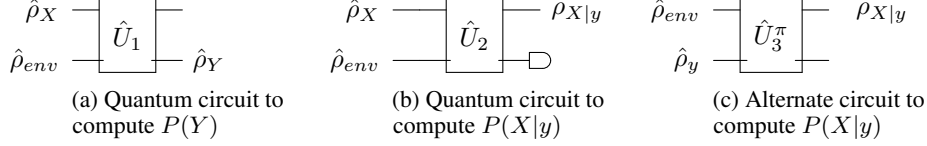

Figure 1: Quantum-circuit analogues of conditioning in graphical models

The quantum circuit for Bayesian update presented by Srinivasan et al. [2018] is shown in Figure 1b. It involves encoding the prior in $\hat{\rho}_X$ as before, and encoding the likelihood table $\mathbf{A}$ in a unitary matrix $\hat{U}_2$. Applying the unitary matrix $\hat{U}_2$ prepares the joint state $\hat{\rho}_{XY}$, and we apply a von Neumann projection operator (denoted $\hat{P}_y$) corresponding to the observation $y$ (the D-shaped symbol in the circuit), to obtain the conditioned state $\hat{\rho}_{X|y}$ in the first particle. The projection operator selects the entries from the joint distribution $\hat{\rho}_{XY}$ that correspond to the actual observation $y$, and zeroes out the other entries, analogous to using an indicator vector to index into a joint probability table. This operation can be written (denominator renormalizes to recover a valid density matrix) as:

$$\hat{\rho}_{X|y} = \frac{\mathrm{tr}_{env}\left(P_y \hat{U}_2 \left(\hat{\rho}_X \otimes \hat{\rho}_{env}\right) \hat{U}_2^\dagger P_y^\dagger\right)}{\mathrm{tr}\left(\mathrm{tr}_{env}\left(P_y \hat{U}_2 \left(\hat{\rho}_X \otimes \hat{\rho}_{env}\right) \hat{U}_2^\dagger P_y^\dagger\right)\right)} \tag{6}$$

However, there is an *alternate* quantum circuit that could implement Bayesian conditioning. Consider re-writing the classical Bayesian update as a linear update as follows:

$$P(X|y) = \left(\mathbf{A} \cdot \mathrm{diag}(\vec{\pi})\right)^T \left(\mathrm{diag}\left(\mathbf{A}\vec{\pi}\right)\right)^{-1} \vec{e}_y \tag{7}$$

where $\left(\mathbf{A} \cdot \mathrm{diag}(\vec{\pi})\right)^T$ yields the joint probability table $P(X, Y)$, $\left(\mathrm{diag}\left(\mathbf{A}\vec{\pi}\right)\right)^{-1}$ is a diagonal matrix with the inverse probabilities $\frac{1}{P(Y=y)}$ on the diagonal, serving to renormalize the columns of the joint probability table $P(X, Y)$. Thus, $\left(\mathbf{A} \cdot \mathrm{diag}(\vec{\pi})\right)^T \left(\mathrm{diag}\left(\mathbf{A}\vec{\pi}\right)\right)^{-1}$ produces a column-stochastic matrix, and $\vec{e}_y$ is just an indicator vector that selects the column corresponding to the observation $y$. Then, just as the circuit in Figure 1a is the quantum generalization for Equation 3, we can use the quantum circuit shown in 1c for this alternate Bayesian update. Here, $\hat{\rho}_y$ encodes the indicator vector corresponding to the observation $y$, and $\hat{U}_3^\pi$ is a unitary matrix constructed using the prior $\pi$ on $X$. Letting $\hat{U}_3^\pi$ to be any unitary matrix constructed from some prior on $X$ would give an alternative quantum Bayesian update.

These are two *different* ways of generalizing classical Bayesian rule within quantum graphical models. So which circuit should we use? One major disadvantage of the second approach is that we must construct different unitary matrices $\hat{U}_3^\pi$ for different priors on $X$. The first approach also explicitly involves measurement, which is nicely analogous to classical observation. As we will see in the next section, the two circuits are different ways of performing inference in Hilbert space, with the first approach being equivalent to Nadaraya-Watson kernel regression and the second approach being equivalent to kernel Bayes rule for Hilbert space embeddings.

## 3 Translating to the language of Hilbert Space Embeddings

In the previous section, we generalized graphical models to quantum graphical models using the quantum view of probability. And since quantum states live in complex Hilbert spaces, inference in QGMs happens in Hilbert space. Here, we re-write the operations on QGMs in the language of Hilbert space embeddings, which should be more familiar to the statistical machine learning community.

### 3.1 Hilbert Space Embeddings

Previous work [Smola et al., 2007] has shown that we can embed probability distributions over a data domain $\mathcal{X}$ in a reproducing kernel Hilbert space (RKHS) $\mathcal{F}$ – a Hilbert space of functions, with some kernel $k$. The feature map $\phi(x) = k(x, \cdot)$ maps data points to the RKHS, and the kernel function satisfies $k(x, x') = \langle \phi(x), \phi(x') \rangle_\mathcal{F}$. Additionally, the dot product in the Hilbert space satisfies the reproducing property:

$$\langle f(\cdot), k(x, \cdot) \rangle_\mathcal{F} = f(x), \quad \text{and} \quad \langle k(x, \cdot), k(x', \cdot) \rangle_\mathcal{F} = k(x, x') \tag{8}$$

### 3.1.1 Mean Embeddings

The following equations describe how a distribution of a random variable $X$ is embedded as a *mean map* [Smola et al., 2007], and how to empirically estimate the mean map from data points $\{x_1, \ldots, x_n\}$ drawn i.i.d from $P(X)$, respectively:

$$\mu_X := \mathbb{E}_X[\phi(X)] \qquad \hat{\mu}_X = \frac{1}{n}\sum_{i=1}^{n}\phi(x_i) \tag{9}$$

**Quantum Mean Maps** We still take the expectation of the features of the data, except we require that the feature maps $\phi(\cdot)$ produce valid density matrices representing pure states (i.e., rank 1). Consequently, quantum mean maps have the nice property of having probabilities along the diagonal. Note that these feature maps can be complex and infinite, and in the latter case, they map to *density operators*. For notational consistency, we require the feature maps to produce rank-1 *vectorized* density matrices (by vertically concatenating the columns of the matrix), and treat the quantum mean map as a vectorized density matrix $\vec{\mu}_X = \vec{\rho}_X$.

### 3.1.2 Cross-Covariance Operators

Cross-covariance operators can be used to embed joint distributions; for example, the joint distribution of random variables $X$ and $Y$ can be represented as a cross-covariance operator (see Song et al. [2013] for more details):

$$\mathcal{C}_{XY} := \mathbb{E}_{XY}[\phi(X) \otimes \phi(Y)] \tag{10}$$

**Quantum Cross-Covariance Operators** The quantum embedding of a joint distribution $P(X, Y)$ is a square $mn \times mn$ density matrix $\hat{\rho}_{XY}$ for constituent $m \times m$ embedding of a sample from $P(X)$ and $n \times n$ embedding of a sample from $P(Y)$. To obtain a quantum cross-covariance matrix $\mathcal{C}_{XY}$, we simply reshape $\hat{\rho}_{XY}$ to an $m^2 \times n^2$ matrix, which is also consistent with estimating it from data as the expectation of outer product of feature maps $\phi(\cdot)$ that produce vectorized density matrices.

### 3.2 Quantum Sum Rule as Kernel Sum Rule

We now re-write the quantum sum rule for quantum graphical models from Equation 4, in the language of Hilbert space embeddings. Srinivasan et al. [2018] showed that Equation 4 can be written as $\hat{\rho}_Y = \sum_i V_i \hat{U} W \hat{\rho}_X W^\dagger \hat{U}^\dagger V_i^\dagger$, where matrices $W$ and $V$ tensor with an environment particle and partial trace respectively. Observe that a quadratic matrix operation can be simplified to a linear operation, i.e., $\hat{U}\hat{\rho}\hat{U}^\dagger = \text{reshape}((\hat{U}^* \otimes \hat{U})\vec{\rho})$ where $\vec{\rho}$ is the vectorized density matrix $\hat{\rho}$. Then:

$$\vec{\mu}_Y = \sum_i \left( \left(V_i \hat{U} W\right)^* \otimes \left(V_i \hat{U} W\right)\right) \vec{\mu}_X = \left(\sum_i \left(V_i \hat{U} W\right)^* \otimes \left(V_i \hat{U} W\right)\right)\vec{\mu}_X = A\vec{\mu}_X \tag{11}$$

where $A = (\sum_i (V_i \hat{U} W)^* \otimes (V_i \hat{U} W))$. We have re-written the complicated transition update as a simple linear operation, though $A$ should have constraints to ensure the operation is valid according to quantum mechanics. Consider estimating $A$ from data by solving a least squares problem: suppose we have data $(\Upsilon_X, \Phi_Y)$ where $\Phi \in \mathbb{R}^{d_1 \times n}, \Upsilon \in \mathbb{R}^{d_2 \times n}$ are matrices of $n$ vectorized $d_1, d_2$-dimensional density matrices and $n$ is the number of data points. Solving for $A$ gives us $A = \Phi_Y \Upsilon_X^\dagger (\Upsilon_X \Upsilon_X^\dagger)^{-1}$. But $\Phi_Y \Upsilon_X^\dagger = n \cdot \mathcal{C}_{YX}$ where $\mathcal{C}_{YX} = \frac{1}{n}\sum_i^n (\vec{\mu}_{Y_i} \otimes \vec{\mu}_{X_i}^\dagger)$. Then, $A = \mathcal{C}_{YX}\mathcal{C}_{XX}^{-1}$, allowing us to re-write Equation 11 as:

$$\vec{\mu}_Y = \mathcal{C}_{YX}\mathcal{C}_{XX}^{-1}\vec{\mu}_X \tag{12}$$

But this is exactly the kernel sum rule from Song et al. [2013], with the conditional embedding operator $\mathcal{C}_{Y|X} = \mathcal{C}_{YX}\mathcal{C}_{XX}^{-1}$. Thus, when the feature maps that produce valid (vectorized) rank-1 density matrices, the quantum sum rule is identical to the kernel sum rule. One thing to note is that solving for $A$ using least-squares needn't preserve the quantum-imposed constraints; so either the learning algorithm must force these constraints, or we project $\vec{\mu}_Y$ back to a valid density matrix.

**Finite Sample Kernel Estimate** We can straightforwardly adopt the kernelized version of the conditional embedding operator from HSEs [Song et al., 2013] ($\lambda$ is a regularization parameter):

$$\mathcal{C}_{Y|X} = \Phi(K_{xx} + \lambda\mathbb{I})^{-1}\Upsilon^\dagger \tag{13}$$

where $\Phi = (\phi(y_1), \ldots, \phi(y_n))$, $\Upsilon = (\phi(x_1), \ldots, \phi(x_n))$, and $K = \Upsilon^\dagger \Upsilon$, and these feature maps produce vectorized rank-1 density matrices. The data points in Hilbert space can be written as $\vec{\mu}_Y = \Phi\alpha_Y$ and $\vec{\mu}_X = \Upsilon\alpha_X$ where $\alpha \in \mathbb{R}^n$ are weights for the training data points, and the kernel quantum sum rule is simply:

$$\vec{\mu}_Y = \mathcal{C}_{Y|X}\vec{\mu}_X \Rightarrow \Phi\alpha_Y = \Phi(K_{xx} + \lambda\mathbb{I})^{-1}\Upsilon^\dagger \Upsilon\alpha_X \Rightarrow \alpha_Y = (K_{xx} + \lambda\mathbb{I})^{-1}K_{xx}\alpha_X \tag{14}$$

### 3.3 Quantum Bayes Rule as Nadaraya-Watson Kernel Regression

Here, we re-write the Bayesian update for QGMs from Equation 6 in the language of HSEs. First, we modify the quantum circuit in 1b to allow for measurement of a rank-1 density matrix $\hat{\rho}_y$ in *any* basis (see Appendix for details) to obtain the circuit shown in Figure 2, described by the equation:

$$\hat{\rho}_{X|y} \propto \text{tr}_{env} \left( (\mathbb{I} \otimes \hat{u}) P(\mathbb{I} \otimes \hat{u}^\dagger) \hat{U} \left( \hat{\rho}_X \otimes \hat{\rho}_{env} \right) \hat{U}^\dagger (\mathbb{I} \otimes \hat{u}^\dagger)^\dagger P^\dagger (\mathbb{I} \otimes \hat{u})^\dagger \right) \tag{15}$$

where $\hat{u}$ changes the basis of the environment variable to one in which the rank-1 density matrix encoding the observation $\hat{\rho}_Y$ is diagonalized to $\Lambda$ – a matrix with all zeros except $\Lambda_{1,1} = 1$. The projection operator will be $P = (\mathbb{I} \otimes \Lambda)$, which means the terms $(\mathbb{I} \otimes \hat{u})P(\mathbb{I} \otimes \hat{u}^\dagger) = (\mathbb{I} \otimes \hat{u})(I \otimes \Lambda)(\mathbb{I} \otimes \hat{u}^\dagger) = (\mathbb{I} \otimes u\Lambda u^\dagger) = (\mathbb{I} \otimes \hat{\rho}_y)$, allowing us to rewrite Equation 15 as:

$$\hat{\rho}_{X|y} \propto \text{tr}_{env} \left( (\mathbb{I} \otimes \hat{\rho}_y) \hat{U} \left( \hat{\rho}_X \otimes \hat{\rho}_{env} \right) \hat{U}^\dagger (\mathbb{I} \otimes \hat{\rho}_y)^\dagger \right) \tag{16}$$

Let us break this equation into two steps:

$$\hat{\rho}_{XY} = \hat{U} \left( \hat{\rho}_X \otimes \hat{\rho}_{env} \right) \hat{U}^\dagger = \hat{U} W \hat{\rho}_X W^\dagger \hat{U}^\dagger$$
$$\hat{\rho}_{X|y} = \frac{\text{tr}_{env} \left( (\mathbb{I} \otimes \hat{\rho}_y) \hat{\rho}_{XY} (\mathbb{I} \otimes \hat{\rho}_y)^\dagger \right)}{\text{tr} \left( \text{tr}_{env} \left( (\mathbb{I} \otimes \hat{\rho}_y) \hat{\rho}_{XY} (\mathbb{I} \otimes \hat{\rho}_y)^\dagger \right) \right)} \tag{17}$$

Now, we re-write the first expression in the language of HSEs. The quadratic matrix operation can be re-written as a linear operation by vectorizing the density matrix as we did in Section 3.2: $\vec{\mu}_{XY} = ((\hat{U}W)^* \otimes (\hat{U}W)) \vec{\mu}_X$. But for $\vec{\mu}_X \in \mathbb{R}^{n^2 \times 1}$, $W \in \mathbb{R}^{ns \times n}$, and $\hat{U} \in \mathbb{C}^{ns \times ns}$ this operation gives $\vec{\mu}_{XY} \in \mathbb{R}^{n^2 s^2 \times 1}$, which we can reshape into an $n^2 \times s^2$ matrix $\mathcal{C}_{XY}^{\pi_X}$ (the superscript simply indicates the matrix was composed from a prior on $X$). We can then directly write $\mathcal{C}_{XY}^{\pi_X} = B \times_3 \vec{\mu}_X$, where $B$ is $((\hat{U}W)^* \otimes (\hat{U}W))$ reshaped into a three-mode tensor and $\times_3$ represents a tensor contraction along the third mode. But, just as we solved $A = \mathcal{C}_{YX} \mathcal{C}_{XX}^{-1}$ in Section 3.2 we can estimate $B = \mathcal{C}_{(XY)X} \mathcal{C}_{XX}^{-1} = \mathcal{C}_{XY|X}$ as a matrix and reshape into a 3-mode tensor, allowing us to re-write the first step in Equation 17 as:

$$\mathcal{C}_{XY}^{\pi_X} = \mathcal{C}_{(XY)X} \mathcal{C}_{XX}^{-1} \vec{\mu}_X = \mathcal{C}_{XY|X} \times_3 \vec{\mu}_X \tag{18}$$

Now, to simplify the second step, observe that the numerator can be rewritten to get $\vec{\mu}_{X|y} \propto \mathcal{C}_{XY}^{\pi_X} \left( \hat{\rho}_y^T \otimes \hat{\rho}_y \right) \vec{t}$, where $\vec{t}$ is a vector of 1s and 0s that carries out the partial trace operation. But, for a rank 1 density matrix $\hat{\rho}_y$, this actually simplifies further:

$$\vec{\mu}_{X|y} \propto \mathcal{C}_{XY}^{\pi_X} \vec{\rho}_y = \left( \mathcal{C}_{XY|X} \times_3 \vec{\mu}_X \right) \vec{\rho}_y \tag{19}$$

One way to renormalize $\vec{\mu}_{X|y}$ is to compute $\left( \mathcal{C}_{XY|X} \times_3 \vec{\mu}_X \right) \vec{\rho}_y$ and reshape it back into a density matrix and divide by its trace. Alternatively, we can rewrite this operation using a vectorized identity matrix $\vec{\mathbb{I}}$ that carries out the full trace in the denominator to renormalize as:

$$\vec{\mu}_{X|y} = \frac{\left( \mathcal{C}_{XY|X} \times_3 \vec{\mu}_X \right) \vec{\rho}_y}{\vec{\mathbb{I}}^T \left( \mathcal{C}_{XY|X} \times_3 \vec{\mu}_X \right) \vec{\rho}_y} \tag{20}$$

**Finite Sample Kernel Estimate**  We kernelize these operations as follows (where $\phi(y) = \vec{\rho}_y$):

$$\vec{\mu}_{X|y} = \frac{\Upsilon \cdot \text{diag} \left( \alpha_X \right) \cdot \Phi^T \phi(y)}{\vec{\mathbb{I}}^T \Upsilon \cdot \text{diag} \left( \alpha_X \right) \cdot \Phi^T \phi(y)} = \frac{\sum_i \Upsilon_i \left( \alpha_X \right)_i k(y_i, y)}{\sum_j \left( \alpha_X \right)_j k(y_j, y)} = \Upsilon \alpha_Y^{(X)} \tag{21}$$

where $(\alpha_Y^{(X)})_i = \frac{(\alpha_X)_i k(y_i, y)}{\sum_j (\alpha_X)_j k(y_j, y)}$, and $\vec{\mathbb{I}}^T \Upsilon = \mathbb{1}^T$ since $\Upsilon$ contains vectorized density matrices, and $\vec{\mathbb{I}}$ carries out the trace operation. As it happens, this method of estimating the conditional embedding

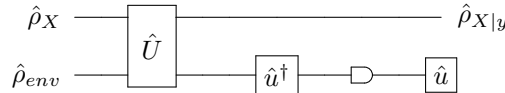

Figure 2: Quantum circuit to compute posterior $P(X|y)$

$\vec{\mu}_{X|y}$ is equivalent to performing Nadaraya-Watson kernel regression [Nadaraya, 1964, Watson, 1964] from the joint embedding to the kernel embedding. Note that this result only holds for the kernels satisfying Equation 4.22 in Wasserman [2006]; importantly, the kernel function must only output positive numbers. One way to enforce this is by using a squared kernel; this is equivalent to a 2$^{\text{nd}}$-order polynomial expansion of the features or computing the outer product of features. Our choice of feature map produces density matrices (as the outer product of features), so their inner product in Hilbert space is equivalent to computing the squared kernel, and this constraint is satisfied.

### 3.4 Quantum Bayes Rule as Kernel Bayes Rule

As we discussed at the end of Section 2.2, Figure 1c is an alternate way of generalizing Bayes rule for QGMs. But following the same approach of rewriting the quantum circuit in the language of Hilbert Space embeddings as in Section 3.2, we get exactly Kernel Bayes Rule [Song et al., 2013]:

$$\vec{\mu}_{X|y} = \mathcal{C}_{XY}^{\pi}(\mathcal{C}_{YY}^{\pi})^{-1}\phi(y) \tag{22}$$

**What we have shown thus far** As promised, we see that the two different but valid ways of generalizing Bayes rule for QGMs affects whether we condition according to Kernel Bayes Rule or Nadaraya-Watson kernel regression. However, we stress that conditioning according to Nadaraya-Watson is computationally much easier; the kernel Bayes rule given by Song et al. [2013] using Gram matrices is written:

$$\hat{\mu}_{X|y} = \Upsilon DK_{yy}((DK_{yy})^2 + \lambda\mathbb{I})^{-1}DK_{:y} \tag{23}$$

where $D = \text{diag}((K_{xx} + \lambda\mathbb{I})^{-1}K_{xx}\alpha_X)$. Observe that this update requires squaring and inverting the Gram matrix $K_{yy}$ – an expensive operation. By contrast, performing Bayesian update using Nadaraya-Watson as per Equation 21 is straightforward. This is one of the key insights of this paper; showing that Nadaraya-Watson kernel regression is an alternate, valid, but simpler way of generalizing Bayes rule to Hilbert space embeddings. We note that interpreting operations on QGMs as inference in Hilbert space is a special case; if the feature maps don't produce density matrices, we can still perform inference in Hilbert space using the quantum/kernel sum rule, and Nadaraya-Watson/kernel Bayes rule, but lose the probabilistic interpretation of a quantum graphical model.

## 4 HSE-HQMMs

We now consider mapping data to vectorized density matrices and modeling the dynamics in Hilbert space using a specific quantum graphical model – hidden quantum Markov models (HQMMs). The quantum circuit for HQMMs is shown in Figure 3 [Srinivasan et al., 2018].

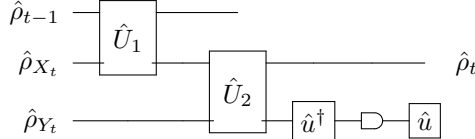

Figure 3: Quantum Circuit for HSE-HQMM

We use the outer product of random Fourier features (RFF) [Rahimi and Recht, 2008] (which produce a valid density matrix) to map data to a Hilbert space. $\hat{U}_1$ encodes transition dynamics, $\hat{U}_2$ encodes observation dynamics, and $\hat{\rho}_t$ is the density matrix after a transition update and conditioning on some observation. The transition and observation equations describing this circuit (with $\hat{U}_I = \mathbb{I} \otimes \hat{u}^{\dagger}$) are:

$$\hat{\rho}'_t = \text{tr}_{\hat{\rho}_{t-1}}\left(\hat{U}_1\left(\hat{\rho}_{t-1} \otimes \hat{\rho}_{X_t}\right)\hat{U}_1^{\dagger}\right) \quad \text{and} \quad \hat{\rho}_t \propto \text{tr}_{Y_t}\left(\hat{U}_I P_y \hat{U}_I^{\dagger}\hat{U}_2\left(\hat{\rho}'_t \otimes \hat{\rho}_{Y_t}\right)\hat{U}_2^{\dagger}\hat{U}_I P_y^{\dagger}\hat{U}_I^{\dagger}\right) \tag{24}$$

As we saw in the previous section, we can rewrite these in the language of Hilbert Space embeddings:

$$\vec{\mu}'_{x_t} = (\mathcal{C}_{x_t x_{t-1}}\mathcal{C}_{x_{t-1}x_{t-1}}^{-1})\vec{\mu}_{x_{t-1}} \quad \text{and} \quad \vec{\mu}_t = \frac{(\mathcal{C}_{x_t y_t|x_t} \times_3 \vec{\mu}'_{x_t})\phi(y_t)}{\mathbb{I}^T(\mathcal{C}_{x_t y_t|x_t} \times_3 \vec{\mu}'_{x_t})\phi(y_t)} \tag{25}$$

And the kernelized version of these operations (where $\Upsilon = (\phi(x_1), \ldots, \phi(x_n))$ is (see appendix):

$$\alpha'_{x_t} = (K_{x_{t-1}x_{t-1}} + \lambda\mathbb{I})^{-1}K_{x_{t-1}x_t}\alpha_{x_{t-1}} \quad \text{and} \quad \alpha_{x_t} = \frac{\sum_i \Upsilon_i\left(\alpha'_{x_t}\right)_i k(y_i, y)}{\sum_j \left(\alpha'_{x_t}\right)_j k(y_j, y)} \tag{26}$$

It is also possible to combine the operations setting $\mathcal{C}_{x_t y_t | x_{t-1}} = \mathcal{C}_{x_t y_t | x_t} \mathcal{C}_{x_t x_{t-1}} \mathcal{C}_{x_{t-1} x_{t-1}}^{-1}$ to write our single update in Hilbert space:

$$\vec{\mu}_{x_t} = \frac{(\mathcal{C}_{x_t y_t | x_{t-1}} \times_3 \vec{\mu}_{x_{t-1}}) \phi(y_t)}{\vec{\mathbb{I}}^T (C_{x_t y_t | x_{t-1}} \times_3 \vec{\mu}_{x_{t-1}}) \phi(y_t)} \tag{27}$$

**Making Predictions** As discussed in Srinivasan et al. [2018], conditioning on some discrete-valued observation $y$ in the quantum model produces an unnormalized density matrix whose trace is the probability of observing $y$. However, in the case of continuous-valued observations, we can go further and treat this trace as the *unnormalized density* of the observation $y_t$, i.e., $f_Y(y_t) \propto \vec{\mathbb{I}}^T (\mathcal{C}_{x_t y_t | x_{t-1}} \times_3 \vec{\mu}'_{t-1}) \phi(y_t)$ – the equivalent operation in the language of quantum circuits is the trace of the unnormalized $\hat{\rho}_t$ shown in Figure 3. A benefit of building this model using the quantum formalism is that we can immediately see that this trace is bounded and lies in $[0, 1]$. It is also straightforward to see that a tighter bound for the unnormalized densities is given by the largest and smallest eigenvalues of the reduced density matrix $\hat{\rho}_{Y_t} = \text{tr}_{X_t}(\hat{\rho}_{X_t Y_t})$ where $\hat{\rho}_{X_t Y_t}$ is the joint density matrix after the application of $\hat{U}_2$.

To make a prediction, we sample from the convex hull of our training set, compute densities as described, and take the expectation to make a point prediction. This formalism is potentially powerful as it lets us maintain a whole distribution over the outputs (e.g. Figure 5), instead of just a point estimate for the next prediction as with LSTMs. A deeper investigation of the density estimation properties of our model would be an interesting direction for future work.

**Learning HSE-HQMMs** We estimate model parameters using 2-stage regression (2SR) [Hefny et al., 2015], and refine them using back-propagation through time (BPTT). With this approach, the learned parameters are not guaranteed to satisfy the quantum constraints, and we handle this by projecting the state back to a valid density matrix at each time step. Details are given in Algorithm 1

---

**Algorithm 1** Learning Algorithm using Two-Stage Regression for HSE-HQMMs

---

**Input:** Data as $\mathbf{y}_{1:T} = \mathbf{y}_1, ..., \mathbf{y}_T$
**Output:** Cross-covariance matrix $\mathcal{C}_{x_t y_t | x_{t-1}}$, can be reshaped into 3-mode tensor for prediction

1: Compute features of the past ($\mathbf{h}$), future ($\mathbf{f}$), shifted future ($\mathbf{s}$) from data (with window $k$):

$$\mathbf{h}_t = h(\mathbf{y}_{t-k:t-1}) \qquad \mathbf{f}_t = f(\mathbf{y}_{t:t+k}) \qquad \mathbf{s}_t = f(\mathbf{y}_{t+1:t+k+1})$$

2: Project data and features of past, future, shifted future into RKHS using random Fourier features of desired kernel (feature map $\phi(\cdot)$) to generate quantum systems:

$$|\mathbf{y}\rangle_t \leftarrow \phi_y(\mathbf{y}_t) \quad |\mathbf{h}\rangle_t \leftarrow \phi_h(\mathbf{h}_t) \quad |\mathbf{f}\rangle_t \leftarrow \phi_f(\mathbf{f}_t) \quad |\mathbf{s}\rangle_t \leftarrow \phi_f(\mathbf{s}_t)$$

3: Construct density matrices in the RKHS and vectorize them:

$$\vec{\rho}_t^{(y)} = \text{vec}\left(|\mathbf{y}\rangle_t \langle \mathbf{y}|_t\right) \quad \vec{\rho}_t^{(h)} = \text{vec}\left(|\mathbf{h}\rangle_t \langle \mathbf{h}|_t\right) \quad \vec{\rho}_t^{(f)} = \text{vec}\left(|\mathbf{f}\rangle_t \langle \mathbf{f}|_t\right) \quad \vec{\rho}_t^{(s)} = \text{vec}\left(|\mathbf{s}\rangle_t \langle \mathbf{s}|_t\right)$$

4: Compose matrices whose columns are the vectorized density matrices, for each available time-step (accounting for window size $k$), denoted $\Phi_y$, $\Upsilon_h$, $\Upsilon_f$, and $\Upsilon_s$ respectively.

5: Obtain extended future via tensor product $\vec{\rho}_t^{(s,y)} \leftarrow \vec{\rho}_t^{(s)} \otimes \vec{\rho}_t^{(y)}$ and collect into matrix $\Gamma_{s,y}$.

6: **Perform Stage 1 regression**

$$\mathcal{C}_{f|h} \leftarrow \Upsilon_f \Upsilon_h^\dagger \left(\Upsilon_h \Upsilon_h^\dagger + \lambda\right)^{-1} \qquad \mathcal{C}_{s,y|h} \leftarrow \Gamma_{s,y} \Upsilon_h^\dagger \left(\Upsilon_h \Upsilon_h^\dagger + \lambda\right)^{-1}$$

7: Use operators from stage 1 regression to obtain denoised predictive quantum states:

$$\tilde{\Upsilon}_{f|h} \leftarrow \mathcal{C}_{f|h} \Upsilon_h \qquad \tilde{\Gamma}_{s,y|h} \leftarrow \mathcal{C}_{s,y|h} \Upsilon_h$$

8: **Perform Stage 2 regression to obtain model parameters**

$$\mathcal{C}_{x_t y_t | x_{t-1}} \leftarrow \tilde{\Gamma}_{s,y|h} \tilde{\Upsilon}_{f|h}^\dagger \left(\tilde{\Upsilon}_{f|h} \tilde{\Upsilon}_{f|h}^\dagger + \lambda\right)^{-1}$$

---

# 5 Comparison with Previous Work

## 5.1 HQMMs

Srinivasan et al. [2018] present a maximum-likelihood learning algorithm to estimate the parameters of a HQMM from data. However, it is very limited in its scope; the algorithm is slow and doesn't scale for large datasets. In this paper, we leverage connections to HSEs, kernel methods, and RFFs to achieve a more practical and scalable learning algorithm for these models. However, one difference to note is that the algorithm presented by Srinivasan et al. [2018] guaranteed that the learned parameters would produce valid quantum operators, whereas our algorithm only approximately produces valid quantum operators; we will need to project the updated state back to the nearest quantum state to ensure that we are tracking a valid quantum system.

## 5.2 PSRNNs

Predictive State Recurrent Neural Networks (PSRNNs) [Downey et al., 2017] are a recent state-of-the-art model developed by embedding a Predictive State Representation (PSR) into an RKHS. The PSRNN update equation is:

$$\vec{\omega}_t = \frac{W \times_3 \vec{\omega}_{t-1} \times_2 \phi(y_t)}{||W \times_3 \vec{\omega}_{t-1} \times_2 \phi(y_t)||_F} \tag{28}$$

where $W$ is a three mode tensor corresponding to the cross-covariance between observations and the state at time $t$ conditioned on the state at time $t-1$, and $\omega$ is a factorization of a p.s.d state matrix $\mu_t = \omega\omega^T$ (so renormalizing $\omega$ by Frobenius norm is equivalent to renormalizing $\mu$ by its trace). There is a clear connection between PSRNNs and the HSE-HQMMs; this matrix $\mu_t$ is what we vectorize to use as our state $\vec{\mu}_t$ in HSE-HQMMs, and both HSE-HQMMs and PSRNNs are parameterized (in the primal space using RFFs) in terms of a three-mode tensor ($W$ for PSRNNs and ($\mathcal{C}_{x_t y_t | x_{t-1}}$ for HSE-HQMMs). We also note that while PSRNNs modified kernel Bayes rule (from Equation 22) *heuristically*, we have shown that this modification can be interpreted as a generalization of Bayes rule for QGMs or Nadaraya-Watson kernel regression. One key difference between these approaches is that we directly use states in Hilbert space to estimate the probability density of observations; in other words HSE-HQMMs are a generative model. By contrast PSRNNs are a discriminative model which rely on an additional ad-hoc mapping from states to observations.

# 6 Experiments

We use the following datasets in our experiments[1]:

- **Penn Tree Bank (PTB)** Marcus et al. [1993]. We train a character-prediction model with a train/test split of 120780/124774 characters due to hardware limitations.

- **Swimmer** Simulated swimmer robot from OpenAI gym[2]. We collect 25 trajectories from a robot that is trained to swim forward (via the cross entropy with a linear policy) with a 20/5 train/test split. There are 5 features at each time step: the angle of the robots nose, together with the 2D angles for each of it's joints.

- **Mocap** Human Motion Capture Dataset. We collect 48 skeletal tracks from three human subjects with a 40/8 train/test split. There are 22 total features at each time step: the 3D positions of the skeletal parts (e.g., upper back, thorax, clavicle).

We compare the performance of three models: HSE-HQMMs, PSRNNs, and LSTMs. We initialize PSRNNs and HSE-HQMMs using Two-Stage Regression (2SR) [Downey et al., 2017] and LSTMs using Xavier Initialization and refine all three models using Back Propagation Through Time (BPTT). We optimize and evaluate all models on Swimmer and Mocap with respect to the Mean Squared Error (MSE) using 10 step predictions as is conventional in the robotics community. This means that to evaluate the model we perform recursive filtering on the test set to produce states, then use these states to make predictions about observations 10 steps in the future. We optimize all models on PTB with respect to Perplexity (Cross Entropy) using 1 step predictions, as is conventional in the NLP community. As we can see in Figure 4, HSE-HQMMs outperform both PSRNNs and LSTMs on the swimmer dataset, and achieve comparable performance to the best alternative on Mocap and PTB. Hyperparameters and other experimental details can be found in Appendix E.

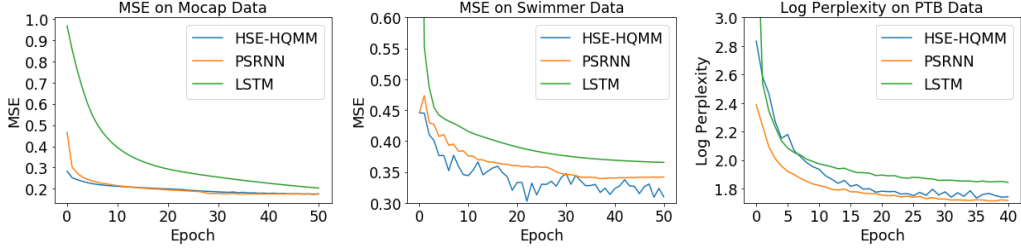

Figure 4: Performance of HSE-HQMM, PSRNN, and LSTM on Mocap, Swimmer, and PTB

**Visualizing Probability Densities** As mentioned previously, HSE-HQMMs can maintain a probability density function over future observations, and we visualize this for a model trained on the Mocap dataset in Figure 5. We take the 22 dimensional joint density and marginalize it to produce three marginal distributions, each over a single feature. We plot the resulting marginal distributions over time using a heatmap, and superimpose the ground-truth and model predictions. We observe that BPTT (second row) improves the marginal distribution. Another interesting observation, from the the last ∼30 timesteps of the marginal distribution in the top-left image, is that our model is able to produce a bi-modal distribution with probability mass at both $y_i = 1.5$ and $y_i = -0.5$, without making any parametric assumptions. This kind of information is difficult to obtain using a discriminative model such as a LSTM or PSRNN. Additional heatmaps can be found in Appendix D.

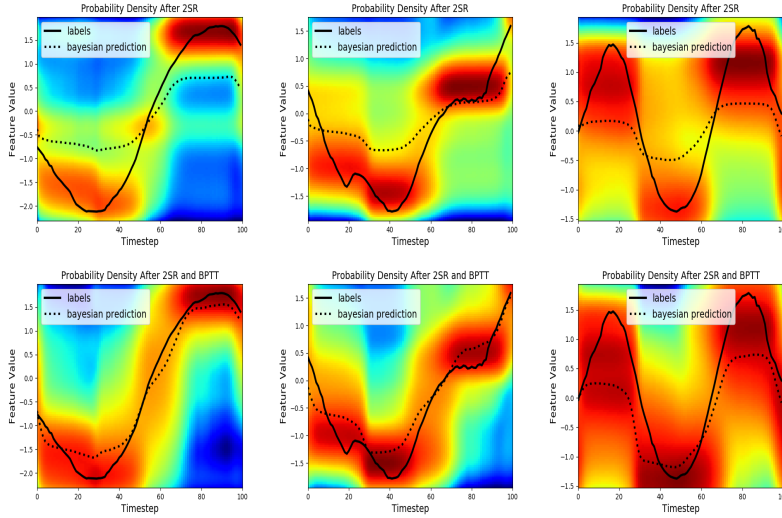

Figure 5: Heatmap Visualizing the Probability Densities generated by our HSE-HQMM model. Red indicates high probability, blue indicates low probability, x-axis corresponds to time, y-axis corresponds to the feature value. Each column corresponds to the predicted marginal distribution of a single feature changing with time. The first row is the probability distribution after 2SR initialization, the second row is the probability distribution after the model in row 1 has been refined via BPTT.

## 7 Conclusion and Future Work

We explored the connections between QGMs and HSEs, and showed that the sum rule and Bayes rule in QGMs is equivalent to kernel sum rule and a special case of Nadaraya-Watson kernel regression. We proposed HSE-HQMMs to model dynamics, and showed experimentally that these models are competitive with LSTMs and PSRNNs on making point predictions, while also being a nonparametric method for maintaining a probability distribution over continuous-valued features. Looking forward, we note that our experiments only consider real kernels/features, so we are not utilizing the full complex Hilbert space; it would be interesting to investigate whether incorporating complex numbers improves our model. Additionally, by estimating parameters using least-squares, the parameters only approximately adhere to quantum constraints. The final model also bears strong resemblance to PSRNNs [Downey et al., 2017]. It would be interesting to investigate both what happens if we are stricter about enforcing quantum constraints, and if we give the model greater freedom to drift from the quantum constraints. Finally, the density estimation properties of the model are also an avenue for future exploration.

## Footnotes

[1]Code will be made available at `https://github.com/cmdowney/hsehqmm`

[2]`https://gym.openai.com/`

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
