[Supplementary Material]

## A    Note on Interpreting Quantum Circuits

Here we briefly describe how to interpret the quantum circuits shown in this paper. Quantum circuits are ubiquitous in the quantum information literature, and we refer the reader to Nielsen and Chuang [2002] for a more complete treatment. We note that we use quantum circuits as a pictorial illustration of the inputs, matrix operations, and outputs in our approach, and these operate on $d$-dimensional quantum states, while traditionally quantum circuits illustrate operations on 2-dimensional quantum states ('qubits').

Generally speaking, the boxes in a quantum circuit represent a bilinear unitary operation (or matrix) on a density matrix, and the wire entering from the left represents the input, and the wire exiting the right represents the output. A matrix $\hat{U}$ that take multiple wires as input/output are operating on a state space that is the tensor product of the state spaces of the inputs. A matrix $\hat{u}$ that takes one wire as input while a parallel wire passes untouched can equivalently be written as a matrix $(\mathbb{I} \otimes \hat{u})$ operating on the joint state space of the two inputs. Finally, note that while we can always tensor two smaller state spaces to obtain a larger state space which we then operate on, the reverse isn't usually possible even though we may draw the wires separately, i.e., it may not be possible to decompose the output of an operation on a larger state space into the smaller state spaces. This is analogous to how distributions of two random variables can be tensored to obtain the joint, but the joint can only be factored if the constituent variables are independent. These examples are illustrated below.

(a) $\hat{\rho}_A \otimes \hat{\rho}_B = \hat{\rho}_{AB}$ if the two states start off separable

(b) $\hat{U}(\hat{\rho}_A \otimes \hat{\rho}_B)\hat{U}^\dagger = \hat{\rho}'_{AB} \neq \hat{\rho}'_A \otimes \hat{\rho}'_b$ in general

(c) $\hat{\rho}_A \otimes \hat{u}\hat{\rho}_B\hat{u}^\dagger = (\mathbb{I} \otimes \hat{u})\rho_{\hat{A}B}(\mathbb{I} \otimes \hat{u})^\dagger$

Figure 6: Simple quantum circuit operations

## B    Modifying the Quantum Circuit for Bayes rule

In Figure 1b, an assumption in the way the observation update is carried out is that the measurement on $\hat{\rho}_Y$ is carried out in the same basis that the unitary operator $\hat{U}$ is performed. When using a HQMM to explicitly model some number of discrete observations, this is fine, since each observation is a basis state, and we always measure in this basis. However, in the general case, we may wish to account for measuring $\hat{\rho}_Y$ in *any* basis. To do this, we get the eigendecomposition of $\hat{\rho}_Y$, so that $u\Lambda u^\dagger = \hat{\rho}_Y$ (the eigenvectors of $\hat{\rho}_Y$ will form a unitary matrix). We only need to rotate $\hat{\rho}_Y$ into the 'correct' basis before measurement; we can leave $\hat{\rho}_{X|y}$ unchanged. This new, general observation is implemented by the circuit in 2. The final $\hat{u}$ is not strictly necessary; since it has no effect on the first particle, which stores $\hat{\rho}_{X|y}$, which is ultimately what we're interested in. However, including it allows us to simplify the update rule.

## C    Kernelizing HSE-HQMMs

Here we derive the kernel embedding update for HSE-HQMMs as given in Equation 26. The observation update follows directly from the Nadaraya-Watson kernel regression given in Equation 21. The transition update comes from a recursive application of the quantum sum rule given in Equation 13. Let $\Upsilon = (\phi(x_{t-1}^{(1)}), \phi(x_{t-1}^{(2)}), \ldots, \phi(x_{t-1}^{(n)}))$ and $\Phi = (\phi(x_t^{(1)}), \phi(x_t^{(2)}), \ldots, \phi(x_t^{(n)}))$. Starting with $\hat{\mu}_{t-1} = \Upsilon\alpha_{t-1}$ using the kernel sum rule recursively twice with $\mathcal{C}_{X_t|X_{t-1}} = \Phi(K_{x_{t-1}x_{t-1}} + \lambda\mathbb{I})^{-1}\Upsilon^\dagger$, we get:

$$\hat{\mu}_{t+1} = \mathcal{C}_{X_t|X_{t-1}}\mathcal{C}_{X_t|X_{t-1}}\hat{\mu}_{t-1}$$
$$\Rightarrow \Phi\alpha_{t+1} = \Phi(K_{x_{t-1}x_{t-1}} + \lambda\mathbb{I})^{-1}\Upsilon^\dagger\Phi(K_{x_{t-1}x_{t-1}} + \lambda\mathbb{I})^{-1}\Upsilon^\dagger\Upsilon\alpha_{t-1} \qquad (29)$$
$$\Rightarrow \alpha_{t+1} = (K_{x_{t-1}x_{t-1}} + \lambda\mathbb{I})^{-1}K_{x_{t-1}x_t}(K_{x_{t-1}x_{t-1}} + \lambda\mathbb{I})^{-1}K_{x_{t-1}x_{t-1}}\alpha_{t-1}$$

If we break apart this two-step update to update for a single time-step, starting from $\alpha_{t-1}$ we can write $\alpha'_t = (K_{x_{t-1}x_{t-1}} + \lambda\mathbb{I})^{-1}K_{x_{t-1}x_t}\alpha_{t-1}$.

# D Additional Heatmaps

Figure 7: Heatmap Visualizing the Probability Densities generated by our HSE-HQMM model on the swimmer dataset. Each column corresponds to single feature. The first row is the probability distribution after 2SR initialization, the second row is the probability distribution after the model in row 1 has been refined via BPTT

# E Experimental Details

## E.1 State Update

Given HSE-HQMM model parameters consisting of the 3-mode tensor $\mathcal{C}_{x_t y_t | x_{t-1}}$, current state $\vec{\mu}_{x_t}$ and observation in Hilbert space $\phi(y_t)$ we perform filtering to update the state as follows:

$$\vec{\mu}_{x_{t+1}} = \left( \mathcal{C}_{x_t y_t | x_{t-1}} \times_3 \vec{\mu}_{x_t} \right) \phi(y_t) \qquad (30)$$

where $A \times_c b$ corresponds to tensor contraction of tensor $A$ with vector $b$ along mode $c$. In our case this means performing tensor contraction of $\mathcal{C}_{x_t y_t | x_{t-1}}$ with the current state and observation along the appropriate modes.

## E.2 Prediction

Given HSE-HQMM model parameters consisting of the 3-mode tensor $\mathcal{C}_{x_t y_t | x_{t-1}}$ and current state $\vec{\mu}_{x_t}$ we can estimate the probability density of an observation $y_t$ (up to normalization) as follows:

$$f_Y(y_t) = \vec{\mathbb{I}}^T ((\mathcal{C}_{x_t y_t | x_{t-1}} \times_3 \mu_{x_t}) \phi(y_t) \qquad (31)$$

If we want to make a point prediction from our model (in order to calculate the mean squared error, etc) we can use the mean $\mathbb{E}[y_t | x_t]$. In practice we approximate this quantity using a sample $y_{1:n}$:

$$\mathbb{E}[y_t | x_t] = \frac{1}{n} \sum_{i=1}^{n} y_i f_Y(y_i) \qquad (32)$$

## E.3 Pure State HSE-HQMMs

In our experiments we use a HSE-HQMM consisting of a single pure state, as opposed to the full density matrix formalism. Using a single pure state is a special case of the general (mixed state) HSE-HQMM, and allows the model to be efficiently implemented. The learning algorithm modified for a pure state is shown in Algorithm 2.

**Algorithm 2** Learning Algorithm using Two-Stage Regression for Pure State HSE-HQMMs

**Input:** Data as $\mathbf{y}_{1:T} = \mathbf{y}_1, ..., \mathbf{y}_T$
**Output:** Cross-covariance matrix $\mathcal{C}_{x_t y_t | x_{t-1}}$, can be reshaped to 3-mode tensor for prediction
1: Compute features of the past ($\mathbf{h}$), future ($\mathbf{f}$), shifted future ($\mathbf{s}$) from data (with window $k$):

$$\mathbf{h}_t = h(\mathbf{y}_{t-k:t-1}) \qquad \mathbf{f}_t = f(\mathbf{y}_{t:t+k}) \qquad \mathbf{s}_t = f(\mathbf{y}_{t+1:t+k+1})$$

2: Project data and features of past, future, shifted future into RKHS using random features of desired kernel (feature map $\phi(\cdot)$) to generate quantum systems:

$$|\mathbf{y}\rangle_t \leftarrow \phi_y(\mathbf{y}_t) \quad |\mathbf{h}\rangle_t \leftarrow \phi_h(\mathbf{h}_t) \quad |\mathbf{f}\rangle_t \leftarrow \phi_f(\mathbf{f}_t) \quad |\mathbf{s}\rangle_t \leftarrow \phi_f(\mathbf{s}_t)$$

3: Compose matrices whose columns are $|\mathbf{y}\rangle_t$, $|\mathbf{h}\rangle_t$, $|\mathbf{f}\rangle_t$, $|\mathbf{s}\rangle_t$ for each available time-step (accounting for window size $k$), denoted $\Phi_y$, $\Upsilon_h$, $\Upsilon_f$, and $\Upsilon_s$ respectively.
4: Obtain extended future via tensor product $|\mathbf{s}, \mathbf{y}\rangle_t \leftarrow |\mathbf{s}\rangle_t \otimes |\mathbf{y}\rangle_t$ and collect into matrix $\Gamma_{s,y}$.
5: **Perform Stage 1 regression**

$$\mathcal{C}_{f|h} \leftarrow \Upsilon_f \Upsilon_h^\dagger \left( \Upsilon_h \Upsilon_h^\dagger + \lambda \right)^{-1}$$

$$\mathcal{C}_{s,y|h} \leftarrow \Gamma_{s,y} \Upsilon_h^\dagger \left( \Upsilon_h \Upsilon_h^\dagger + \lambda \right)^{-1}$$

6: Use operators from stage 1 regression to obtain denoised predictive quantum states:

$$\tilde{\Upsilon}_{f|h} \leftarrow \mathcal{C}_{f|h} \Upsilon_h$$

$$\tilde{\Gamma}_{s,y|h} \leftarrow \mathcal{C}_{s,y|h} \Upsilon_h$$

7: **Perform Stage 2 regression to obtain model parameters**

$$\mathcal{C}_{x_t y_t | x_{t-1}} \leftarrow \tilde{\Gamma}_{s,y|h} \tilde{\Upsilon}_{f|h}^\dagger \left( \tilde{\Upsilon}_{f|h} \tilde{\Upsilon}_{f|h}^\dagger + \lambda \right)^{-1}$$

---

### E.4 Parameter Values

Hyperparameter settings can be found in table 1. Some additional details: In two-stage regression we use history (future) features consisting of the past (next) 10 observations as one-hot vectors concatenated together. We use a ridge-regression parameter of $0.05$ (this is consistent with the values suggested in Boots et al. [2013], Downey et al. [2017]). The kernel width is set to the median pairwise (Euclidean) distance between neighboring data points. We use a fixed learning rate of $0.1$ for BPTT with a BPTT horizon of 20.

| Parameter | Value |
|---|---:|
| Kernel | Gaussian |
| Kernel Width | Median Neighboring Pairwise Distance |
| Number of Random Fourier Features | 1000 |
| Prediction Horizon | 10 |
| Batch Size | 20 |
| State Size | 20 |
| Max Gradient Norm | 0.25 |
| Number of Layers | 1 |
| Ridge Regression Regularizer | 0.05 |
| Learning Rate | 0.1 |
| Learning algorithm | Stochastic Gradient Descent (SGD) |
| Number of Epochs | 50 |
| BPTT Horizon | 20 |

Table 1: Hyperparameter values used in experiments