[Reviews · NeurIPS 2018]

Reviewer 1



The paper develops a connection between quantum graphical models and inference with Hilbert space embeddings (HSE). It shows kernel formulations for sum rule and Bayes rule with quantum graphical models, connecting kernel mean embeddings as well as cross-covariance operators. The paper proposes to use HSE on top of HQMMs, arguing that it improves empirical performance against PSRNN and LSTMs. The paper is nicely written and does a very good job at explaining difficult quantum concepts using tools that should be familiar to a machine learning audience. I personally really enjoyed reading and learning about the connections developed in the paper. I also think the work is very original and could lead to interesting follow up work. I have a few questions regarding the experiments: - Why not compare with a simpler baseline, such as regular HMM? - There are plenty of existing standard datasets for sequence prediction / modeling, yet none of the experiments in this paper seem to use any of them, which makes it harder to convince others that the improvement is not because of weak baselines and unusual train/test splits. - What are the per iteration BPTT efficiency and modeling complexity for HSE-HQMM compared with other baselines, namely LSTM and PSRNN? - I am rather confused by the PTB experiments. The log-perplexity for all these methods are no more than 2, which means that the perplexity is below 10. However, the current SOTA for PTB perplexity should be significantly higher (in the 30-50 range). Minor points: Heat map resolution and aspect ratio. Typo on line 106: embeddingsx ========== Thanks for the clarifications!

Reviewer 2



The paper "Using Quantum Graphical Models to Perform Inference in Hilbert Space" provides several results related to linking QGMs and HSEs, including the equivalence between the sum rule for QGMs and kernel sum rule for HSEs. The paper further introduces HSE-HQMM to model dynamics, and empirically shows its improvement over LSTMs and PSRNNs. Overall, the paper is well written with important results in it. From what I understand, the technical details are correct, but it is also possible that I misunderstand some critical steps since this is not my research area. One minor suggestion is that the authors can improve the quality of their Figures 4 and 5 such as making fonts larger, unifying the fonts and making the figure captions more informative. I have read other reviews and authors' response. I would like to keep my original scores.

Reviewer 3



This paper establishes correspondences between quantum graphical models and Hilbert space embeddings. Based on these correspondences, the authors propose a new model (HSE-HQMM) as a nonparametric approach to model probability distribution and make predictions. Experimental results demonstrate better performance for the proposed method over existing methods. Specific comments follow: 1. an explanation of why quantum-circuits are needed in the presentation will be helpful. 2. It will also be helpful to provide some details of the hardware specification and runtime of the experiments. 3. More applications other than predictions will be great, given the versatility of HSE-HQMM. ===== The reviewer would like to thank the authors for their explanation.